# A Game Theoretic Model of Adversaries and Media Manipulation

**Kjell Hausken** 

Faculty of Science and Technology, University of Stavanger, 4036 Stavanger, Norway; kjell.hausken@uis.no

**Abstract:** A model is developed for two players exerting media manipulation efforts to support each of two actors who interact controversially. Early evidence may support one actor, while the full evidence emerging later may support the other actor. Exerting effort when the full evidence exceeds (falls short off) the early evidence is rewarded (punished) with lower (higher) unit effort cost. Properties and simulations are presented to illustrate the players' strategic challenges when altering eight model parameters, i.e., a player's unit effort cost, stake in the interaction, proportionality parameter scaling the strength of reward or punishment, time discount parameter, early evidence, full evidence, contest intensity, and evidence ratio intensity. Realizing the logic of the model may aid understanding on how to handle the difference between early and full evidence of controversies, in which players have an ideological stake.

**Keywords:** media; game; adversaries; players; contest; manipulation; spin control

## 1. Introduction

In 1983, 90% of the American media was owned by 50 companies, while in 2011 only six companies (General Electric, News Corp, Disney, Viacom, Time Warner, and CBS) owned the same 90% of the American media [1]. Since the year 1900s, independent news media has become subject to more concentrated control. With fewer and more powerful players, ideological impact becomes a more prominent concern. Economies of scale, and possibly other factors, may enable individual players to impose their ideological views more fiercely, without being compromised by a plurality of multifarious other players. Reporting objectively, truthfully, and with ideologically neutrality, is challenging. Tribe [2] suggests that tools in policy science are themselves ideologically biased. Levins [3] and Nagy, Fairbrother, Etterson, and Orme-Zavaleta [4] suggest that truth may emerge by intersecting independent lies. (That is, various independent models may together resemble truth.)

One widely reported controversial media scenario, which led to subsequent lawsuits, was between high school student Nicholas Sandmann from northern Kentucky Covington Catholic High School and the 64-year old native American Nathan Phillips at the Lincoln Memorial in Washington D.C., USA [5,6]. Sandmann wore a Make America Great Again hat and was in a group of fellow students on an annual school trip to attend the pro-life March for Life, combined with sightseeing. They waited for a bus to Kentucky. Phillips was beating a drum and chanting, while partaking in a group of Native American marchers attending the Indigenous Peoples March. Phillips was locking eyes with a smiling Sandmann a few inches apart. Early selective videos of the interaction on 18 January 2019 led most of the media to criticize Sandmann and the students for potentially provoking Phillips. It turned out that many videos and audio recordings of the interaction existed given the presence of many people at the prominent location. As accumulated and more full evidence of the interaction emerged, a view gradually arose that, potentially, the story was the opposite of that originally reported, and that Phillips was potentially provoking Sandmann. This remarkable turn of events caused the media to subsequently react in many different manners, hypothetically, in accordance with their ideological position. Examples of the

media's reactions, from one extreme to the other, were to retract and apologize, rewrite, reinterpret due to new evidence, ignore, retain the original account with some rewriting, or retain the original account with no adjustment.

Motivated by the potential ideological concerns of media organizations, and scenarios, such as the one sketched above, this article develops a model of the background where two adversarial actors interact controversially. The model is technically a one-period model, but accounts for early evidence of the interaction emerging in period 1, and full accumulated evidence emerging in period 2. Before period 1 two adversarial actors interact controversially. Incomplete early evidence emerges. Two media organizations are the players. Each player supports one of the actors ideologically, and exerts manipulation efforts including spin control to persuade media consumers that the actor he represents is righteous and should not be blamed. Media manipulation effort is interpreted broadly to include competition, verbal fighting, etc. Hirshleifer [7] interprets fighting as a metaphor, i.e., "falling also into the category of interference struggles are political campaigns, rent-seeking maneuvers for licenses and monopoly privileges [8], commercial efforts to raise rivals' costs [9], strikes and lockouts, and litigation—all being conflictual activities that need not involve actual violence." The early and full evidence, and a variety of characteristics of the two players are incorporated into the game they play.

Hardly any literature exists on the phenomenon. Allcott and Gentzkow [10] analyze fake news and social media in the 2016 US election. Blom and Hansen [11], Zannettou, Chatzis, Papadamou, and Sirivianos [12], and Khoja [13] consider clickbait news. Kshetri and Voas [14] examine fake news within an economic perspective.

The article is organized as follows. Section 2 presents the model. Section 3 analyzes the model. Section 4 illustrates with an example. Section 5 concludes.

## 2. The model

Appendix A shows the nomenclature. Consider two actors 1 and 2 which are adversaries interacting controversially. An actor may be an individual, a group, or any collective unit. Player *i* is a media media organization which reports on both actors and their interaction, while identifying with actor *i*, $i = 1, 2$, see Figure 1.

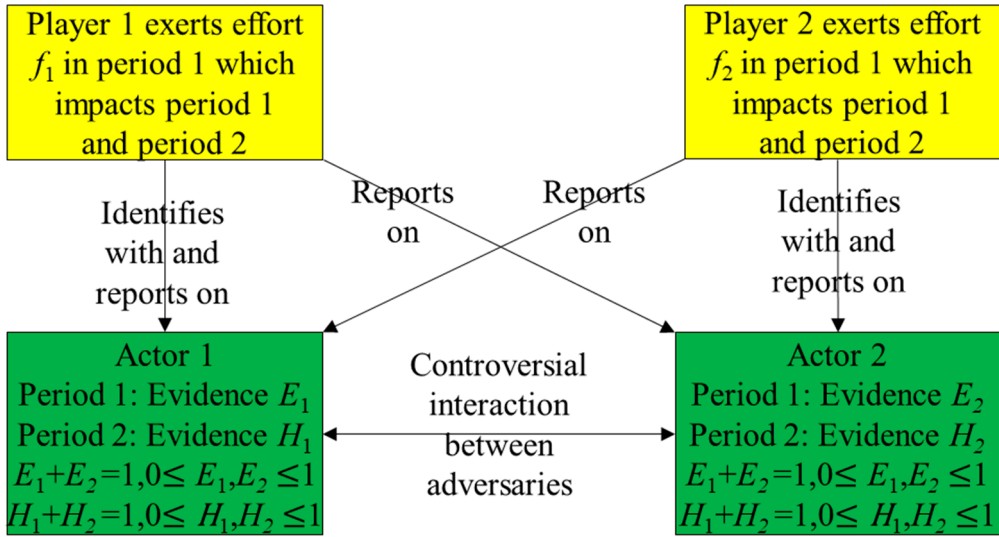

**Figure 1.** Two adversaries actor 1 and actor 2 interacting controversially, reported on by player 1 (media organization 1) which identifies with actor 1, and player 2 (media organization 2), which identifies with actor 2.

## 2.1. Period 1

Assume that some early evidence $E_i$, $0 \leq E_i \leq 1$, $E_1 + E_2 = 1$, is available before period 1 supporting player $i$ and actor $i$. Early evidence $E_i$ may be preliminary videos and audios of the interaction, neutral witness accounts, etc. Early evidence $E_i$ may be incomplete and capture only part of the truth, or may be framed so that it reports a falsehood, e.g. if the beginning and end of a video are deleted. Player $i$ in period 1 exerts media manipulation effort $f_i$ at unit cost $a_i$ to manipulate the perception of early evidence $E_i$ so that it is perceived to be higher. Effort $f_i$ is player $i$'s only strategic choice variable.

The two players (i.e., the two media organizations) are in a contest modeled with the common ratio form contest success function [15]. Player $i$ earns a proportion of the stake, expressing the degree to which actor $i$ is perceived not to be at fault in the interaction. Player $i$ has a probability $p_i$ of success, $0 \leq p_i \leq 1$, $p_1 + p_2 = 1$. Analogous to Hirshleifer and Osborne's [16] Litigation Success Function, [1] where we apply early evidence $E_i$ instead of truth (which is unavailable), we consider the Media Manipulation Success Function ratio,

$$\frac{p_1}{p_2} = \left(\frac{f_1}{f_2}\right)^{\alpha}\left(\frac{E_1}{E_2}\right)^{\gamma} \tag{1}$$

where $\alpha$, $\alpha \geq 0$, is the contest intensity, and $\gamma$, $\gamma \geq 0$, is the early evidence ratio intensity with an interpretation analogous to $\alpha$. The right-hand side of (1) contains the media manipulation effort ratio $f_1/f_2$ raised to $\alpha$, and the early evidence ratio $E_1/E_2$ raised to $\gamma$. When $\alpha = 0$ or $f_1 = f_2$, the success ratio $p_1/p_2$ equals the early evidence ratio $E_1/E_2$ raised to $\gamma$. When $0 \leq \alpha \leq 1$, exerting less effort $f_i$ than the other player has disproportional advantage. When $\alpha = 1$, exerting effort $f_i$ has proportional advantage. When $\alpha > 1$, exerting more effort $f_i$ than the other player has disproportional advantage. When $\alpha = \infty$, exerting slightly more effort $f_i$ than the other player gives a "winner-takes-all" situation.

Since $E_1$ and $E_2$ are parameters, $\left(\frac{E_1}{E_2}\right)^{\gamma}$ is also a parameter where the early evidence ratio intensity $\gamma$ depends on culture, law, the nature of the interaction, the time and place of the interaction, the identities of actors 1 and 2 and players 1 and 2, and how the early evidence $E_i$ is generated, presented, and framed. Equation (1) is rewritten as a contest success probability,

$$p_i = \frac{f_i^{\alpha} E_i^{\gamma}}{f_1^{\alpha} E_1^{\gamma} + f_2^{\alpha} E_2^{\gamma}} \tag{2}$$

for player $i$. Assuming that player $i$ has a stake $J_i$ in the interaction, his expected utility in period 1 is,

$$U_i = p_i J_i - a_i f_i = \frac{f_i^{\alpha} E_i^{\gamma}}{f_1^{\alpha} E_1^{\gamma} + f_2^{\alpha} E_2^{\gamma}} J_i - a_i f_i \tag{3}$$

where (2) has been inserted.

## 2.2. Period 2

The early evidence $E_i$ in period 1 is incomplete. Assume that additional evidence becomes available in period 2, causing the accumulated or full evidence $H_i$ supporting player $i$, $0 \leq H_i \leq 1$, $H_1 + H_2 = 1$. The full evidence $H_i$ in period 2 includes early evidence $E_i$ from period 1, and additional evidence in period 2. Thus $H_i = E_i$ means either that no new evidence becomes available in period 2, or that the new available evidence in period 2 does not alter how the evidence impacts the support of players 1 and 2. In contrast, $H_i \neq E_i$ means more ($H_i > E_i$) or less ($H_i < E_i$) evidence supporting player $i$ in period 2. The full evidence $H_i$ may e.g., include the beginning and end of the video missing in

---

[1]　See Hausken, Levitin, and Levitin [17] for an application of the contest success function to lawsuits, and Hausken and Levitin [18] for an application of the contest success function to risk analysis.

the period 1 early evidence $E_i$, or additional videos or audio. Although, the full evidence $H_i$ becomes available in period 2, we assume that the players make no strategic choices in period 2.

Modeling the full evidence $H_i$ in player $i$'s expected utility for period 2 involves three characteristics and no strategic choice variables for period 2. First, since $H_i > E_i$ means that more evidence becomes available, supporting player $i$, player $i$ should be rewarded proportionally to $H_i - E_i$. Since player $i$ was disadvantaged with low early evidence $E_i$ in the contest success function in (3), higher full evidence $H_i$ should advantage player $i$. In contrast, since $H_i < E_i$ means that the full evidence $H_i$ supports player $i$ to a lower extent, player $i$ should be punished proportionally to $H_i - E_i$. Second, a proportionality parameter $Q_i$, $Q_i \geq 0$, is introduced to scale the strength of the reward or punishment $H_i - E_i$. Third, proportionality with player $i$'s effort $f_i$ is assumed since the reward or punishment $H_i - E_i$ should be higher (lower) if player $i$'s effort $f_i$ is higher (lower). Letting these three characteristics operate multiplicatively, player $i$'s utility in period 2 is:

$$V_i = -Q_i(E_i - H_i)f_i. \tag{4}$$

The full evidence $H_i$ becomes available after early evidence $E_i$. The time duration may be a few days, a few seconds or many years, depending on the phenomenon. Hence a time discount parameter $\delta_i$, $0 \leq \delta_i \leq 1$, is introduced, where $\delta_i = 0$ means no emphasis on the future so that the full evidence $H_i$ is irrelevant, and $\delta_i = 1$ means that the future is as important as the present. Letting the future be more important than the present is possible, but probably more uncommon. In other words, as in Equation (5) by Hausken [19], assume that player $i$ has time discount parameter $\delta_i$, $0 \leq \delta_i \leq 1$. Consequently, player $i$'s expected utility over the two periods is,

$$W_i = U_i + \delta_i V_i = \frac{f_i^\alpha E_i^\gamma}{f_1^\alpha E_1^\gamma + f_2^\alpha E_2^\gamma} J_i - (a_i + \delta_i Q_i(E_i - H_i))f_i. \tag{5}$$

where $U_i$ and $V_i$ are inserted from (3) and (4). The term $a_i + \delta_i Q_i(E_i - H_i)$ can be interpreted as player $i$'s actual unit effort cost (accounting for the reward or punishment) when $a_i + \delta_i Q_i(E_i - H_i) \geq 0$, and as player $i$'s actual unit effort benefit (accounting for substantial reward) when $a_i + \delta_i Q_i(E_i - H_i) \leq 0$. The latter unit effort benefit is uncommon, and arises when $H_i \geq E_i + \frac{a_i}{\delta_i Q_i}$, which means that the accumulated evidence $H_i$ supporting player $i$ in period 2 overwhelms the early evidence $E_i$ supporting player $i$ in period 1. This uncommon event causes infinite effort $f_i$. The following Assumption 1 excludes this uncommon event: Assumption 1: $H_i \leq E_i + \frac{a_i}{\delta_i Q_i}$.

## 3. Analyzing the Model

Since the players make no strategic choices in period 2, the game is technically a one-period game with strategic choice variables $f_1$ and $f_2$ in period 1. Differentiating player $i$'s expected utility in (5) with respect to his free choice variable $f_i$ in period 1, and equating with zero, gives:

$$\begin{aligned} \frac{\partial W_1}{\partial f_1} &= \frac{\alpha E_1^\gamma E_2^\gamma f_1^{\alpha-1} f_2^\alpha}{\left(f_1^\alpha E_1^\gamma + f_2^\alpha E_2^\gamma\right)^2} J_1 - (a_1 + \delta_1 Q_1(E_1 - H_1)) = 0, \\ \frac{\partial W_2}{\partial f_2} &= \frac{\alpha E_1^\gamma E_2^\gamma f_1^\alpha f_2^{\alpha-1}}{\left(f_1^\alpha E_1^\gamma + f_2^\alpha E_2^\gamma\right)^2} J_2 - (a_2 + \delta_2 Q_2(E_2 - H_2)) = 0. \end{aligned} \tag{6}$$

Solving (6) when Assumption 1 is satisfied gives:

$$f_1 = \frac{J_1(a_2 + \delta_2 Q_2(E_2 - H_2))}{J_2(a_1 + \delta_1 Q_1(E_1 - H_1))} f_2, \quad f_2 = \frac{\frac{\alpha E_2^\gamma J_2^{1+\alpha}(a_1 + \delta_1 Q_1(E_1 - H_1))^\alpha}{E_1^\gamma J_1^\alpha (a_2 + \delta_2 Q_2(E_2 - H_2))^{1+\alpha}}}{\left(1 + \frac{E_2^\gamma J_2^\alpha(a_1 + \delta_1 Q_1(E_1 - H_1))^\alpha}{E_1^\gamma J_1^\alpha(a_2 + \delta_2 Q_2(E_2 - H_2))^\alpha}\right)^2}. \tag{7}$$

The second order derivatives, inserting (7), are satisfied as negative, i.e.,

$$
\begin{aligned}
\frac{\partial^2 W_1}{\partial f_1^2} &= -\frac{(a_1+\delta_1 Q_1(E_1-H_1))^{2-\alpha}}{\alpha E_1^\gamma E_2^\gamma J_1^{1+\alpha} J_2^\alpha (a_2+\delta_2 Q_2(E_2-H_2))^\alpha} \\
&\times \left( E_2^\gamma J_2^\alpha (a_1+\delta_1 Q_1(E_1-H_1))^\alpha + E_1^\gamma J_1^\alpha (a_2+\delta_2 Q_2(E_2-H_2))^\alpha \right) \\
&\times \left( E_2^\gamma J_2^\alpha (1-\alpha)(a_1+\delta_1 Q_1(E_1-H_1))^\alpha + E_1^\gamma J_1^\alpha (1+\alpha)(a_2+\delta_2 Q_2(E_2-H_2))^\alpha \right), \\
\frac{\partial^2 W_2}{\partial f_2^2} &= -\frac{(a_2+\delta_2 Q_2(E_2-H_2))^{2-\alpha}}{\alpha E_1^\gamma E_2^\gamma J_1^\alpha J_2^{1+\alpha} (a_1+\delta_1 Q_1(E_1-H_1))^\alpha} \\
&\times \left( E_2^\gamma J_2^\alpha (a_1+\delta_1 Q_1(E_1-H_1))^\alpha + E_1^\gamma J_1^\alpha (a_2+\delta_2 Q_2(E_2-H_2))^\alpha \right) \\
&\times \left( E_2^\gamma J_2^\alpha (1+\alpha)(a_1+\delta_1 Q_1(E_1-H_1))^\alpha + E_1^\gamma J_1^\alpha (1-\alpha)(a_2+\delta_2 Q_2(E_2-H_2))^\alpha \right)
\end{aligned}
\tag{8}
$$

Inserting (7) into (5), when Assumption 1 is satisfied, gives the player's expected utilities:

$$
W_1 = \frac{\left( 1+(1-\alpha)\frac{E_2^\gamma J_2^\alpha (a_1+\delta_1 Q_1(E_1-H_1))^\alpha}{E_1^\gamma J_1^\alpha (a_2+\delta_2 Q_2(E_2-H_2))^\alpha} \right) J_1}{\left( 1+\frac{E_2^\gamma J_2^\alpha (a_1+\delta_1 Q_1(E_1-H_1))^\alpha}{E_1^\gamma J_1^\alpha (a_2+\delta_2 Q_2(E_2-H_2))^\alpha} \right)^2},
$$

$$
W_2 = \frac{\frac{E_2^\gamma J_2^\alpha (a_1+\delta_1 Q_1(E_1-H_1))^\alpha}{E_1^\gamma J_1^\alpha (a_2+\delta_2 Q_2(E_2-H_2))^\alpha} \left( 1-\alpha+\frac{E_2^\gamma J_2^\alpha (a_1+\delta_1 Q_1(E_1-H_1))^\alpha}{E_1^\gamma J_1^\alpha (a_2+\delta_2 Q_2(E_2-H_2))^\alpha} \right) J_2}{\left( 1+\frac{E_2^\gamma J_2^\alpha (a_1+\delta_1 Q_1(E_1-H_1))^\alpha}{E_1^\gamma J_1^\alpha (a_2+\delta_2 Q_2(E_2-H_2))^\alpha} \right)^2}
\tag{9}
$$

which are positive when,

$$
\alpha \le 1 + Min\left\{ \frac{E_1^\gamma J_1^\alpha (a_2+\delta_2 Q_2(E_2-H_2))^\alpha}{E_2^\gamma J_2^\alpha (a_1+\delta_1 Q_1(E_1-H_1))^\alpha}, \frac{E_2^\gamma J_2^\alpha (a_1+\delta_1 Q_1(E_1-H_1))^\alpha}{E_1^\gamma J_1^\alpha (a_2+\delta_2 Q_2(E_2-H_2))^\alpha} \right\}
\tag{10}
$$

which is a lenient restriction on the contest intensity $\alpha$. For equivalent players (10) simplifies to $\alpha \le 2$. When (10) is not satisfied, assume without loss of generality that $\frac{E_1^\gamma J_1^\alpha (a_2+\delta_2 Q_2(E_2-H_2))^\alpha}{E_2^\gamma J_2^\alpha (a_1+\delta_1 Q_1(E_1-H_1))^\alpha} \ge 1$, which means that player 2 is disadvantaged e.g., due to a higher unit effort cost $a_2$. To avoid negative expected utility, player 2 exerts zero effort $f_2$ earning zero expected utility $W_2$. If player 2 chooses zero effort $f_2$, player 1 cannot choose arbitrarily low but positive effort, $f_1$, which would not be an equilibrium, since player 2 would deviate by choosing some positive effort. Hence, we assume that player 1 chooses an effort $f_1$ which deters player 2 from receiving positive expected utility $W_2$. This is accomplished by solving player 1's first order condition $\frac{\partial W_1}{\partial f_1} = 0$, i.e., the first Equation in (6), together with $W_2 = 0$ determined by (5). These are two equations with two unknown $f_1$ and $f_2$ which are solved to yield,

$$
f_1 = \left( \frac{f_2^\alpha E_2^\gamma}{E_1^\gamma} \left( \frac{J_2}{(a_2+\delta_2 Q_2(E_2-H_2))f_2} - 1 \right) \right)^{1/\alpha}
\tag{11}
$$

which cannot be solved analytically, but is illustrated numerically in the next section.

Property 1. For the interior solution when Assumption 1 is satisfied, $\alpha = 1$, $i = 1,2$, $i \ne j$, $\frac{\partial f_i}{\partial a_i} \le 0$, $\frac{\partial f_j}{\partial a_i} \le 0$ when $E_i^\gamma J_i\left( a_j+\delta_j Q_j\left(E_j-H_j\right)\right) \le E_j^\gamma J_j(a_i+\delta_i Q_i(E_i-H_i))$, $\frac{\partial W_i}{\partial a_i} \le 0$, $\frac{\partial W_j}{\partial a_i} \ge 0$. Proof. Appendix B Equations (A1), (A2), (A3), (A4).

Intuitively, a higher unit effort cost $a_i$ in period 1 discourages player $i$ causing lower effort $f_i$ and lower expected utility $W_i$. In contrast, higher $a_i$ causes higher expected utility $W_j$ for player $j$, and lower effort $f_j$ when the specified inequality is satisfied. The specified inequality is more easily satisfied when $a_i$, $J_j$ and $H_j$ are high, which advantage player $j$.

**Property 2.** For the interior solution when Assumption 1 is satisfied, $\alpha = 1$, $i = 1, 2$, $i \neq j$, $\frac{\partial f_i}{\partial J_i} \geq 0$, $\frac{\partial f_j}{\partial J_i} \geq 0$ when $E_i^\gamma J_i\left(a_j + \delta_j Q_j \left(E_j - H_j\right)\right) \leq E_j^\gamma J_j(a_i + \delta_i Q_i(E_i - H_i))$, $\frac{\partial W_i}{\partial J_i} \geq 0$, $\frac{\partial W_j}{\partial J_i} \leq 0$. Proof. Appendix B Equations (A5), (A6), (A7), (A8).

Higher stake $J_i$ in the interaction in period 1 for player $i$ induces higher effort $f_i$ and higher expected utility $W_i$. In contrast, higher $J_i$ causes lower expected utility $W_j$ for player $j$, and higher effort $f_j$ when the same inequality as in Property 1 is satisfied.

**Property 3.** For the interior solution when Assumption 1 is satisfied, $\alpha = 1$, $i = 1, 2$, $i \neq j$, $\frac{\partial f_i}{\partial Q_i} \geq 0$ when $E_i \leq H_i$, $\frac{\partial f_j}{\partial Q_i} \geq 0$ when $E_i \leq H_i$ and $E_i^\gamma J_i\left(a_j + \delta_j Q_j\left(E_j - H_j\right)\right) \leq E_j^\gamma J_j(a_i + \delta_i Q_i(E_i - H_i))$, $\frac{\partial W_i}{\partial Q_i} \geq 0$ when $E_i \leq H_i$, $\frac{\partial W_j}{\partial Q_i} \leq 0$ when $E_i \leq H_i$. Proof. Appendix B Equations (A9), (A10), (A11), (A12).

The impact of player $i$'s proportionality parameter $Q_i$ depends on whether the early evidence $E_i$ supporting player $i$ in period 1 is lower or higher than the accumulated evidence $H_i$ supporting player $i$ in period 2. When $E_i \leq H_i$, so that player $i$ benefits from transitioning from period 1 to period 2, higher $Q_i$ induces player $i$ to exert higher effort $f_i$ and he receives higher expected utility $W_i$. In contrast, player $j$ receives lower expected utility $W_j$, and exerts higher effort $f_j$ when the same inequality as in Properties 1 and 2 is satisfied.

**Property 4.** For the interior solution when Assumption 1 is satisfied, $\alpha = 1$, $i = 1, 2$, $i \neq j$, $\frac{\partial f_i}{\partial \delta_i} \geq 0$ when $E_i \leq H_i$, $\frac{\partial f_j}{\partial \delta_i} \geq 0$ when $E_i \leq H_i$ and $E_i^\gamma J_i\left(a_j + \delta_j Q_j\left(E_j - H_j\right)\right) \leq E_j^\gamma J_j(a_i + \delta_i Q_i(E_i - H_i))$, $\frac{\partial W_i}{\partial \delta_i} \geq 0$ when $E_i \leq H_i$, $\frac{\partial W_j}{\partial \delta_i} \leq 0$ when $E_i \leq H_i$. Proof. Appendix B Equations (A13), (A14), (A15), (A16).

The impact of player $i$'s time discount parameter $\delta_i$ is equivalent to the impact of the proportionality parameter $Q_i$, except that $\delta_i$ is confined to the interval $0 \leq \delta_i \leq 1$, while $Q_i$ is unbounded from above, i.e., $Q_i \geq 0$.

**Property 5.** For the interior solution when Assumption 1 is satisfied, $\alpha = \gamma = 1$, $i = 1, 2$, $i \neq j$, $\frac{\partial f_i}{\partial E_i} \geq 0$ when $E_i^\gamma J_i\left(a_j + \delta_j Q_j\left(E_j - H_j\right)\right) \leq E_j^\gamma J_j(a_i - \delta_i Q_i(E_i + H_i))$, $\frac{\partial f_j}{\partial E_i} \geq 0$ when $a_i \geq \delta_i Q_i H_i$ and $E_i^\gamma J_i\left(a_j + \delta_j Q_j\left(E_j - H_j\right)\right) \leq E_j^\gamma J_j(a_i + \delta_i Q_i(E_i - H_i))$, $\frac{\partial W_i}{\partial E_i} \geq 0$ when $a_i \geq \delta_i Q_i H_i$, $\frac{\partial W_j}{\partial E_i} \leq 0$ when $a_i \geq \delta_i Q_i H_i$. Proof. Appendix B Equations (A17), (A18), (A19), (A20).

Property 5 assumes the intermediate value $\gamma = 1$ for the early evidence ratio intensity in period 1, to simplify the analysis. More early evidence $E_i$ supporting player $i$ in period 1 causes higher expected utility $W_i$ for player $i$ and lower expected utility $W_j$ for player $j$ when $a_i \geq \delta_i Q_i H_i$, i.e., when player $i$'s unit effort cost $a_i$ is high compared with his discount parameter $\delta_i$, proportionality parameter $Q_i$, and the accumulated evidence $H_i$ supporting him in period 2. More early evidence $E_i$ supporting player $i$ in period 1 causes higher effort $f_i$ for player $i$ when the same inequality as in Properties 1,2,3,4 is satisfied. Higher $E_i$ also causes higher effort $f_j$ for player $j$ when player $i$ is disadvantaged with $a_i \geq \delta_i Q_i H_i$, and the same inequality as in Properties 1,2,3,4 is satisfied.

**Property 6.** For the interior solution when Assumption 1 is satisfied, $\alpha = \gamma = 1$, $i = 1, 2$, $i \neq j$, $\frac{\partial f_i}{\partial H_i} \geq 0$, $\frac{\partial f_j}{\partial H_i} \geq 0$ when $E_i^\gamma J_i\left(a_j + \delta_j Q_j\left(E_j - H_j\right)\right) \leq E_j^\gamma J_j(a_i + \delta_i Q_i(E_i - H_i))$, $\frac{\partial W_i}{\partial H_i} \geq 0$, $\frac{\partial W_j}{\partial H_i} \leq 0$. Proof. Appendix B Equations (A21), (A22), (A23), (A24).

Property 6 also assumes the intermediate value $\gamma = 1$ for the early evidence ratio intensity in period 1, to simplify the analysis. Again, and intuitively, more accumulated evidence $H_i$ supporting player $i$ in period 2 causes higher effort $f_i$ expected utility $W_i$ for player $i$ and lower expected utility $W_j$ for player $j$. Higher $H_i$ causes higher effort $f_j$ for player $j$ when same inequality as in Properties 1,2,3,4,5 is satisfied.

**Property 7.** For the interior solution when Assumption 1 is satisfied, $i = 1, 2$, $i \neq j$, $\frac{\partial f_i}{\partial \alpha} \geq 0$ and $\frac{\partial f_j}{\partial \alpha} \geq 0$ when $E_1^\gamma J_1^\alpha (a_2 + \delta_2 Q_2 (E_2 - H_2))^\alpha \left(1 + \alpha Ln\left(\frac{J_2(a_1 + \delta_1 Q_1(E_1 - H_1))}{J_1(a_2 + \delta_2 Q_2(E_2 - H_2))}\right)\right) + E_2^\gamma J_2^\alpha (a_1 + \delta_1 Q_1(E_1 - H_1))^\alpha$ $\left(1 + \alpha Ln\left(\frac{J_1(a_2 + \delta_2 Q_2(E_2 - H_2))}{J_2(a_1 + \delta_1 Q_1(E_1 - H_1))}\right)\right)$, $\frac{\partial W_i}{\partial \alpha} \geq 0$ when $E_1^\gamma J_1^\alpha (a_2 + \delta_2 Q_2(E_2 - H_2))^\alpha + E_2^\gamma J_2^\alpha (a_1 + \delta_1 Q_1(E_1 - H_1))^\alpha$ $+ \left(E_1^\gamma J_1^\alpha (1 + \alpha)(a_2 + \delta_2 Q_2(E_2 - H_2))^\alpha + E_2^\gamma J_2^\alpha (1 - \alpha)(a_1 + \delta_1 Q_1(E_1 - H_1))^\alpha\right) Ln\left(\frac{J_2(a_1 + \delta_1 Q_1(E_1 - H_1))}{J_1(a_2 + \delta_2 Q_2(E_2 - H_2))}\right) \leq 0$,

$$\frac{\partial W_j}{\partial \alpha} \quad \leq \quad 0 \quad \text{when} \quad E_1^\gamma J_1^\alpha (a_2 + \delta_2 Q_2 (E_2 - H_2))^\alpha + E_2^\gamma J_2^\alpha (a_1 + \delta_1 Q_1 (E_1 - H_1))^\alpha$$

$$+ \left( E_1^\gamma J_1^\alpha (1 - \alpha)(a_2 + \delta_2 Q_2 (E_2 - H_2))^\alpha + E_2^\gamma J_2^\alpha (1 + \alpha)(a_1 + \delta_1 Q_1 (E_1 - H_1))^\alpha \right) \quad Ln \left( \frac{J_1(a_2 + \delta_2 Q_2 (E_2 - H_2))}{J_2(a_1 + \delta_1 Q_1 (E_1 - H_1))} \right) \geq 0.$$

Proof. Appendix B Equations (A25), (A26), (A27), (A28).

Property 8. For the interior solution when Assumption 1 is satisfied, $\alpha = 1$, $i = 1, 2$, $i \neq j$, $\frac{\partial f_i}{\partial \gamma} \leq 0$ and $\frac{\partial f_j}{\partial \gamma} \leq 0$ when $Ln(E_1) \leq Ln(E_2)$ and $E_i^\gamma J_i \left( a_j + \delta_j Q_j \left( E_j - H_j \right) \right) \leq E_j^\gamma J_j (a_i + \delta_i Q_i (E_i - H_i))$, $\frac{\partial W_i}{\partial \gamma} \leq 0$ and $\frac{\partial W_j}{\partial \gamma} \geq 0$ when $Ln(E_1) \leq Ln(E_2)$. Proof. Appendix B Equations (A29), (A30), (A31), (A32).

## 4. Illustrating the Solution

Referring to the scenario in the introduction, in this section we can think of actor 1 as Catholic Kentucky high school student, Nicholas Sandmann, and actor 2 as native American, Nathan Phillips. We can think of player 1 as the parts of the media that supported or identified ideologically with Nicholas Sandmann. Possible examples are the Covington Catholic High School newspaper and local media institutions in Covington, Kentucky, or various catholic media outlets. We can think of player 2 as the parts of the media that supported or identified ideologically with Nathan Phillips. Possible examples are native American media outlets, and the media institutions that Sandmann filed lawsuits against.

Figure 2 illustrates the solution with the benchmark parameter values $a_i = J_i = Q_i = \alpha = \gamma = \delta_i = 1$, $E_1 = 0.1$, $H_1 = 0.8$, which imply $E_2 = 0.9$, $H_2 = 0.2$, $i = 1, 2$. That is, actor 1 supported by player 1 is assigned substantial fault expressed as low $E_1 = 0.1$ in period 1, and much less fault expressed as low $H_1 = 0.8$ in period 2, and vice versa for actor 2 supported by player 2. That causes low actual unit effort cost $a_1 + \delta_1 Q_1 (E_1 - H_1) = 0.3$ for player 1, and high actual unit effort cost $a_2 + \delta_2 Q_2 (E_2 - H_2) = 1.7$ for player 2, at the benchmark. Player 2 nevertheless receives the highest expected utility $W_2 > W_1$ at the benchmark since the early evidence ratio $E_1 / E_2 = 1/9$ favors player 2 in the benchmark contest. In each of the eight panels one parameter value varies, while the other parameter values are kept at their benchmarks.

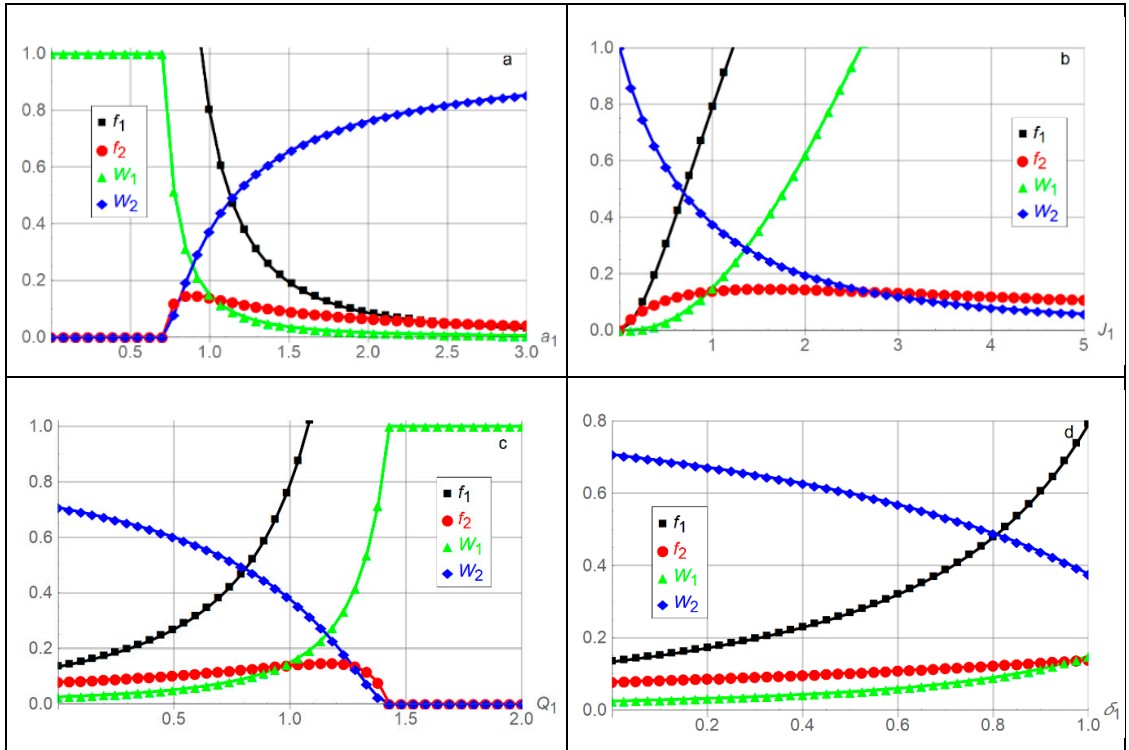

**Figure 2.** *Cont.*

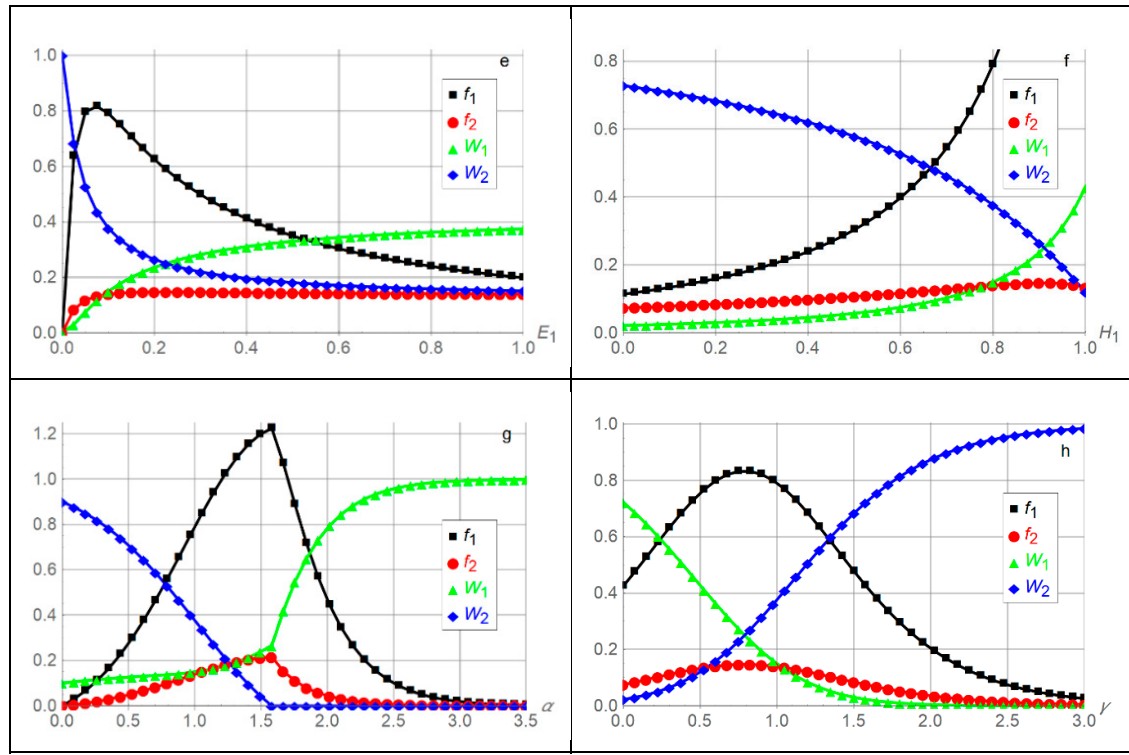

**Figure 2.** Media manipulation efforts $f_1$ and $f_2$ and expected utilities $W_1$ and $W_2$ for players 1 and 2 as functions of $a_1$, $J_1$, $Q_1$, $\delta_1$, $E_1$, $H_1$, $\alpha$, and $\gamma$ relative to the benchmark parameter values $a_i = J_i = Q_i = \alpha = \gamma = \delta_i = 1$, $E_1 = 0.1$, $H_1 = 0.8$, which imply $E_2 = 0.9$, $H_2 = 0.2$, $i = 1, 2$

In Figure 2 panel a, a high unit effort cost $a_1$ for player 1 causes low effort $f_1$ and low expected utility $W_1$ for player 1, $\lim_{a_1 \infty} f_1 = \lim_{a_1 \infty} W_1 = 0$, and high expected utility $\lim_{a_1 \infty} W_2 = J_2 = 1$ for player 2. Player 2 is advantaged when $a_1$ is high, which does not induce a need to exert high effort $f_2$, and $\lim_{a_1 \infty} f_2 = 0$. As $a_1$ decreases, the players have equal unit effort costs when $a_1 = a_2 = 1$. Then, player 1 exerts high effort $f_1$ and receives low expected utility $W_1$, and vice versa, player 2 exerts low effort $f_2$ and receives higher expected utility $W_2$. Although, player 1 is rewarded with a low actual unit effort cost 0.3 at the benchmark, the contest is not sufficiently beneficial for player 1. That is, player 2 is advantaged at the benchmark. As $a_1$ decreases below 1 to $a_1 = 0.88$, the players receive equal expected utilities $W_1 = W_2 = 0.25$. Decreasing $a_1$ further to $a_1 = 0.7$ causes player 1's actual unit effort cost to be $a_1 + \delta_1 Q_1 (E_1 - H_1) = 0$, enabling player 1 to exert arbitrarily high effort $f_1$ at no cost. Hence for $a_1 \leq 0.7$, player 1 receives expected utility $W_1 = J_1 = 1$, and player 2 exerts effort $f_2 = 0$ and receives expected utility $W_2 = 0$.

In Figure 2 panel b, increasing stake $J_1$ in the interaction in period 1 for player 1 causes increasing effort $f_1$ and increasing expected utility $W_1$ for player 1. From (7), $\lim_{J_1 \infty} f_1 = \frac{0.9}{0.17} \approx 5.29$, see Appendix C Figure A1. From (9), $\lim_{J_1 \infty} W_1 = \infty$. In contrast, player 2's effort $f_2$ is inverse U shaped as $J_1$ increases. This common phenomenon arises because player 2 is advantaged when $J_1$ is low, receiving high expected utility $W_2$, and disadvantaged when $J_1$ is high, receiving low expected utility $W_2$. Inserting into (7) and (9), $\lim_{J_1 \infty} f_2 = \lim_{J_1 \infty} W_2 = 0$. If $J_1 = J_2$, rather than $J_1$ varies along the horizontal axis in panel b, $\lim_{J_1 \infty} f_1 = \lim_{J_1 \infty} W_1 = \lim_{J_1 \infty} f_2 = \lim_{J_1 \infty} W_2 = \infty$, where player 2 is advantaged throughout since player 2 is advantaged at the benchmark $J_1 = J_2 = 1$.

In Figure 2 panel c, decreasing the proportionality parameter $Q_1$ which scales the strength of the reward $H_1 - E_1 = 0.7$ for player 1 has an impact similar to increasing player 1's unit effort cost $a_1$ in Figure 2 panel a, due to the opposite roles $Q_1$ and $a_1$ play in player 1's actual unit effort cost $a_1 + \delta_1 Q_1 (E_1 - H_1)$ when $E_1 - H_1 = -0.7$ is negative. Hence when $Q_1$ is low, $f_1$, $W_1$, and $f_2$ are low, and

player 2's expected utility $W_2$ is high. As the proportionality parameter $Q_1$ increases, player 1 benefits from the lower actual unit effort cost which causes higher effort $f_1$ and higher expected utility $W_1$ for player 1, and lower expected utility $W_2$ for player 2. As in Figure 2 panels a and b, player 2's effort $f_2$ is inverse U shaped due to being advantaged when $Q_1$ is low and disadvantaged when $Q_1$ is high. When $Q_1$ increases to $Q_1 = 1.43$, player 1's actual unit effort cost decreases to $a_1 + \delta_1 Q_1 (E_1 - H_1) = 0$, enabling player 1 to exert arbitrarily high effort $f_1$ at no cost. Hence for $Q_1 \geq 1.43$, player 1 receives expected utility $W_1 = J_1 = 1$, and player 2 exerts effort $f_2 = 0$ and receives expected utility $W_2 = 0$.

In Figure 2 panel d, varying player 1's time discount parameter $\delta_1$ between 0 and 1 when $Q_1 = 1$ is equivalent to varying player 1's proportionality parameter $Q_1$ between 0 and 1 when $\delta_1 = 1$, since $\delta_1$ and $Q_1$ only occur as $\delta_1 Q_1$. Figure 2 panel d highlights how disadvantaged player 1 becomes by being shortsighted expressed with low $\delta_1$. That is, player 1 does not envision the reward flowing from $E_1 - H_1 = -0.7$ being negative, exerts low effort $f_1$ and receives low expected utility $W_1$. In contrast, player 2, endowed with the benchmark discount parameter $\delta_2 = 1$, benefits from player 1's shortsightedness when $\delta_1$ is low, and receives high expected utility $W_2$.

In Figure 2 panel e, decreasing player 1's early evidence $E_1$ supporting player 1 in period 1 below the already low benchmark $E_1 = 0.1$ increases his reward $H_1 - E_1$, but decreases the evidence ratio $E_1 / E_2$ in (1) and (5). Player 1 gets less incentive to conduct media manipulation effort since $E_1 / E_2$ is low, but can manipulate the media more cheaply since his actual unit effort cost $a_1 + \delta_1 Q_1 (E_1 - H_1)$ is low. Hence decreasing $E_1$ below $E_1 = 0.1$ causes player 1's effort $f_1$ to be inverse U-shaped and eventually decrease towards zero, while his expected utility $W_1$ decreases to zero, and player 2's expected utility increases to $W_2 = J_2 = 1$ when $E_1 = 0$. In contrast, increasing player 1's early evidence $E_1$ above his benchmark $E_1 = 0.1$ decreases his reward $H_1 - E_1$, causing his actual unit effort cost $a_1 + \delta_1 Q_1 (E_1 - H_1)$ to increase. When $E_1 > H_1$, the reward becomes a punishment, since the early evidence $E_1$ supporting player 1 in period 1 is higher than the accumulated evidence $H_1$ supporting player 1 in period 2. That gives a higher actual unit effort cost causing player 1's effort $f_1$ to decrease. As $E_1$ increases, player 2's effort $f_2$ is slightly inverse U shaped, player 1's expected utility $W_1$ increases, and player 2's expected utility $W_2$ decreases.

In Figure 2 panel f, increasing player 1's accumulated evidence $H_1$ supporting player 1 in period 2 above the already high benchmark $H_1 = 0.8$, increases his reward $H_1 - E_1$, which decreases his actual unit effort cost $a_1 + \delta_1 Q_1 (E_1 - H_1)$. Hence his effort $f_1$ increases, reaching $f_1 = 2.26$ when $H_1 = 1$ (outside what is plotted in panel f), and his expected utility $W_1$ increases. As $H_1$ increases, player 2's effort $f_2$ is slightly inverse U shaped, and his expected utility $W_2$ decreases. In contrast, decreasing player 1's accumulated evidence $H_1$ below his benchmark $H_1 = 0.8$ decreases his reward $H_1 - E_1$, causing his actual unit effort cost $a_1 + \delta_1 Q_1 (E_1 - H_1)$ to increase. Hence, his effort $f_1$ and expected utility $W_1$ decrease, while player 2's effort $f_2$ decreases and his expected utility $W_2$ increases. As $H_1$ decreases below $E_1 = 0.1$, player 1's actual unit effort cost increases above his unit effort cost $a_1$ since $H_1 - E_1$ becomes a punishment.

In Figure 2 panel g, decreasing the contest intensity $\alpha$ below the benchmark $\alpha = 1$ causes player 1 to exert lower effort $f_1$ and receive lower expected utility $W_1$. A lower $\alpha$ causes efforts $f_1$ and $f_2$ to have lower impact on the contest, which becomes more egalitarian, and 100% egalitarian with no impact on the contest when $\alpha = 0$. Hence, player 1's advantage of a lower actual unit effort cost 0.3 than 1.7 for player 2 at the benchmark gradually gets eroded. Hence, lower $\alpha$ causes higher expected utility $W_2$ for player 2, sustained by decreasing effort $f_2$. In contrast, increasing $\alpha$ above the benchmark $\alpha = 1$ causes higher effort $f_1$ and expected utility $W_1$ for player 1, and higher effort $f_2$ and lower expected utility $W_2$ for player 2, up to when $\alpha = 1.58$. Higher contest intensity $\alpha$ is usually characterized by both players exerting higher efforts, which is costly. The efforts $f_1$ and $f_2$ cannot increase without bounds. At some point the weakest player reaches his limit. Thus player 2's expected utility is $W_2 = 0$ when $\alpha \geq 1.58$. Player 2 may exert zero effort or some positive effort $f_2$ when $\alpha \geq 1.58$, as long as his expected utility $W_2$ is not negative. As discussed in the previous section, if player 2 chooses zero effort $f_2$, player 1 cannot choose an arbitrarily low, but positive effort $f_1$, which would not be an equilibrium,

since player 2 would deviate by choosing some positive effort. Applying (11), Figure 2 panel g for $\alpha \geq 1.58$ is determined numerically. Continuous efforts $f_1$ and $f_2$ are ensured through $\alpha = 1.58$. As $\alpha$ increases above $\alpha = 1.58$, decreasing effort $f_1$ by player 1 suffices to deter player 2 from exerting effort $f_2$ to obtain positive expected utility $W_2$. Thus player 1's expected utility $W_1$ increases concavely, $\lim_{\alpha\infty} W_1 = J_1 = 1$, while $\lim_{\alpha\infty} f_1 = \lim_{\alpha\infty} f_2 = \lim_{\alpha\infty} W_2 = 0$.

In Figure 2 panel h, increasing the early evidence ratio intensity $\gamma$ from zero is beneficial for player 2, $\lim_{\gamma\infty} W_2 = J_2 = 1$, and not beneficial for player 1, $\lim_{\gamma\infty} W_1 = 0$, accompanied by $\lim_{\gamma\infty} f_1 = \lim_{\gamma\infty} f_2 = 0$. To see this, inserting the benchmark parameter values when $\gamma$ varies into (7) and (9) gives,

$$f_1 = \frac{17}{3} f_2, \; f_2 = \frac{0.3 \times 9^\gamma}{1.7^2 \left(1 + \frac{3\times 9^\gamma}{17}\right)^2}, W_1 = \frac{1}{\left(1 + \frac{3\times 9^\gamma}{17}\right)^2}, W_2 = \frac{\left(\frac{3\times 9^\gamma}{17}\right)^2}{\left(1 + \frac{3\times 9^\gamma}{17}\right)^2} \tag{12}$$

where the inverse early evidence ratio $E_2/E_1 = 9$ raised to $\gamma$, i.e., $(E_2/E_1)^\gamma = 9^\gamma$, favors player 2 in terms of higher expected utility $W_2 > W_1$ at the benchmark when $\gamma = 1$. Player 2 is favored increasingly when $\gamma > 1$, and decreasingly when $\gamma < 1$. Both players' efforts $f_1$ and $f_2$ are inverse U shaped in $\gamma$ since one player is advantaged when the other is disadvantaged, and vice versa, with the highest media manipulation efforts $f_1$ and $f_2$ for intermediate $\gamma$. In other words, for low early evidence ratio intensity, the fact that player 1 is subject to low early evidence is ameliorated, his effort matters less, and he receives high expected utility. In contrast, high early evidence ratio intensity amplifies how player 1 is subject to low early evidence, giving his lower expected utility.

## 5. Conclusions

A model is developed for two adversarial actors, which interact controversially. Early incomplete evidence emerges about which actor is at fault. A game is analyzed between two media organizations, as the players identifying ideologically with each of the two actors. Each player exerts manipulation efforts to support the actor he represents in a contest with the other player. We consider the two actors by comparison with the scenario in the introduction and simulation section, as high school student Nicholas Sandmann and native American Nathan Phillips, who interacted controversially January 18, 2019 at the Lincoln Memorial in Washington D.C., USA. We can think of the two players as two media organizations, which try to report the facts from the interaction, but additionally have ideological or other preferences that induce them to report favorably on the actor they identify with and support.

The game is technically a one-period game, where each player exerts one effort, but accounts for early evidence emerging in period 1 and full evidence emerging in period 2. If the full evidence equals the early evidence, each player's unit effort cost has a fixed value. If the full evidence supports an actor more (less) than the early evidence, the player identifying with that actor is rewarded (punished) with a lower (higher) unit effort cost proportional to the strength of the additional (decreased) support and proportional to a time discount parameter. The article illustrates each player's strategic challenge in determining the amount of media manipulation effort to exert, while accounting for the difference between the early and full evidence, the unit cost, and various other parameters.

To specify the model's implications, properties are developed for the model's eight parameters, which are illustrated with simulations relative to a benchmark. Without the loss of generality, actor 1 is supported by player 1 and is assumed to be substantially at fault, based on the early evidence, and much less at fault based on the full evidence. The impacts of player 1's unit effort cost and stake in the interaction are discussed. Higher proportionality parameter, scaling the strength of the reward to player 1 for being disadvantaged with low early evidence, and higher time discount parameter for player 1, cause higher effort and expected utility for player 1, and inverse U shaped effort and lower expected utility for player 2. Inverse U shapes are common when one player decreases his effort when either, advantaged or disadvantaged, and exerts high effort when being neither, advantaged nor disadvantaged. Increasing the early evidence supporting player 1 from zero causes inverse U

shaped efforts for both players, increasing expected utility for player 1, and decreasing expected utility for player 2. Increasing the full evidence supporting player 1 from zero causes increasing effort and expected utility for player 1, and decreasing expected utility for player 2. Increasing the contest intensity with the given benchmark is shown to increase both players' efforts until a point where the disadvantaged player is deterred and receives zero expected utility. One implication for scenarios, such as the one between Nicholas Sandmann and Nathan Phillips, which gained widespread coverage and some degree of intensity, is that media organizations supporting one of the actors may potentially deter, outcompete, or silence the opposing media organizations. Finally, by increasing early evidence ratio intensity with the given benchmark demonstrates a common example where both players' efforts are inverse U shaped, player 1's expected utility decreases, and player 2's expected utility increases. The prevalence of inverse *U* shaped results illustrates how the interaction between media organizations as players may often be characterized by one player or the other being advantaged, which may compromise, objectively, the neutral and ideology-free reporting.

The article provides a tool for media organizations, analysts, consumers, regulators, researchers, and regular people to better understand how adversarial interaction may play out in today's continuously evolving media landscape. Realizing the interests of each player and actor, how each player and actor interact, and how new information becomes available over time, as illustrated in this article, may potentially enable everyone involved to contribute to mutually beneficial future media development. Future research should apply the model to more than two adversarial actors, more than two media players, more than two time periods, different kinds of information, and incorporate the role of media owners, regulators, advertisers, and consumers.

**Author Contributions:** The author contributed everything to this article by himself.

**Funding:** No funding was received.

**Acknowledgments:** I thank Jun Zhuang for suggesting an analysis of adversaries and media manipulation, and useful discussions in the early development of the article.

**Conflicts of Interest:** No conflicts of interest exist.

## Appendix A. Nomenclature

*Parameters*

| | |
|---|---|
| $a_i$ | Player $i$'s unit effort cost in period 1, $i = 1, 2$ |
| $Q_i$ | Proportionality parameter which scales the strength of reward or punishment $E_i - H_i$, $i = 1, 2$ |
| $J_i$ | Player $i$'s stake in the interaction in period 1, $i = 1, 2$ |
| $\delta_i$ | Player $i$'s time discount parameter, $i = 1, 2$ |
| $E_i$ | Early evidence supporting player $i$ in period 1, $i = 1, 2$ |
| $H_i$ | Accumulated evidence supporting player $i$ in period 2, $i = 1, 2$ |
| $\alpha$ | Contest intensity in period 1 |
| $\gamma$ | Early evidence ratio intensity in period 1 |

*Strategic choice variable*

| | |
|---|---|
| $f_i$ | Player $i$'s media manipulation effort in period 1, $i = 1, 2$ |

*Dependent variables*

| | |
|---|---|
| $p_i$ | Player $i$'s probability of success in period 1, $i = 1, 2$ |
| $U_i$ | Player $i$'s expected utility in period 1, $i = 1, 2$ |
| $V_i$ | Player $i$'s utility in period 2, $i = 1, 2$ |
| $W_i$ | Player $i$'s expected utility over the two periods, $i = 1, 2$ |

## Appendix B. Proof of Properties 1–8

$$\frac{\partial f_1}{\partial a_1} = -\frac{2E_1^{\gamma}E_2^{2\gamma}J_1^2 J_2^2 (a_2 + \delta_2 Q_2 (E_2 - H_2))}{\left(E_1^{\gamma} J_1 (a_2 + \delta_2 Q_2 (E_2 - H_2)) + E_2^{\gamma} J_2 (a_1 + \delta_1 Q_1 (E_1 - H_1))\right)^3} \tag{A1}$$

$$\frac{\partial f_2}{\partial a_1} = \frac{E_1^{\gamma}E_2^{\gamma}J_1 J_2^2 \left(E_1^{\gamma} J_1 (a_2 + \delta_2 Q_2 (E_2 - H_2)) - E_2^{\gamma} J_2 (a_1 + \delta_1 Q_1 (E_1 - H_1))\right)}{\left(E_1^{\gamma} J_1 (a_2 + \delta_2 Q_2 (E_2 - H_2)) + E_2^{\gamma} J_2 (a_1 + \delta_1 Q_1 (E_1 - H_1))\right)^3} \tag{A2}$$

$$\frac{\partial W_1}{\partial a_1} = -\frac{2E_1^{2\gamma}E_2^{\gamma}J_1^3 J_2 (a_2 + \delta_2 Q_2 (E_2 - H_2))^2}{\left(E_1^{\gamma} J_1 (a_2 + \delta_2 Q_2 (E_2 - H_2)) + E_2^{\gamma} J_2 (a_1 + \delta_1 Q_1 (E_1 - H_1))\right)^3} \tag{A3}$$

$$\frac{\partial W_2}{\partial a_1} = \frac{2E_1^{\gamma}E_2^{2\gamma}J_1 J_2^3 (a_1 + \delta_1 Q_1 (E_1 - H_1))(a_2 + \delta_2 Q_2 (E_2 - H_2))}{\left(E_1^{\gamma} J_1 (a_2 + \delta_2 Q_2 (E_2 - H_2)) + E_2^{\gamma} J_2 (a_1 + \delta_1 Q_1 (E_1 - H_1))\right)^3} \tag{A4}$$

$$\frac{\partial f_1}{\partial J_1} = \frac{2E_1^{\gamma}E_2^{2\gamma}J_1 J_2^2 (a_1 + \delta_1 Q_1 (E_1 - H_1))(a_2 + \delta_2 Q_2 (E_2 - H_2))}{\left(E_1^{\gamma} J_1 (a_2 + \delta_2 Q_2 (E_2 - H_2)) + E_2^{\gamma} J_2 (a_1 + \delta_1 Q_1 (E_1 - H_1))\right)^3} \tag{A5}$$

$$\frac{\partial f_2}{\partial J_1} = \frac{E_1^{\gamma}E_2^{\gamma}J_2^2 (a_1 + \delta_1 Q_1 (E_1 - H_1))\left(E_2^{\gamma} J_2 (a_1 + \delta_1 Q_1 (E_1 - H_1)) - E_1^{\gamma} J_1 (a_2 + \delta_2 Q_2 (E_2 - H_2))\right)}{\left(E_1^{\gamma} J_1 (a_2 + \delta_2 Q_2 (E_2 - H_2)) + E_2^{\gamma} J_2 (a_1 + \delta_1 Q_1 (E_1 - H_1))\right)^3} \tag{A6}$$

$$\frac{\partial W_1}{\partial J_1} = \frac{E_1^{2\gamma}J_1^2 (a_2 + \delta_2 Q_2 (E_2 - H_2))^2 \left(3E_2^{\gamma} J_2 (a_1 + \delta_1 Q_1 (E_1 - H_1)) + E_1^{\gamma} J_1 (a_2 + \delta_2 Q_2 (E_2 - H_2))\right)}{\left(E_1^{\gamma} J_1 (a_2 + \delta_2 Q_2 (E_2 - H_2)) + E_2^{\gamma} J_2 (a_1 + \delta_1 Q_1 (E_1 - H_1))\right)^3} \tag{A7}$$

$$\frac{\partial W_2}{\partial J_1} = -\frac{2E_1^{\gamma}E_2^{2\gamma}J_2^3 (a_1 + \delta_1 Q_1 (E_1 - H_1))^2 (a_2 + \delta_2 Q_2 (E_2 - H_2))}{\left(E_1^{\gamma} J_1 (a_2 + \delta_2 Q_2 (E_2 - H_2)) + E_2^{\gamma} J_2 (a_1 + \delta_1 Q_1 (E_1 - H_1))\right)^3} \tag{A8}$$

$$\frac{\partial f_1}{\partial Q_1} = -\frac{2E_1^{\gamma}E_2^{2\gamma}J_1^2 J_2^2 \delta_1 (E_1 - H_1)(a_2 + \delta_2 Q_2 (E_2 - H_2))}{\left(E_1^{\gamma} J_1 (a_2 + \delta_2 Q_2 (E_2 - H_2)) + E_2^{\gamma} J_2 (a_1 + \delta_1 Q_1 (E_1 - H_1))\right)^3} \tag{A9}$$

$$\frac{\partial f_2}{\partial Q_1} = \frac{E_1^{\gamma}E_2^{\gamma}J_1 J_2^2 \delta_1 (E_1 - H_1)\left(E_1^{\gamma} J_1 (a_2 + \delta_2 Q_2 (E_2 - H_2)) - E_2^{\gamma} J_2 (a_1 + \delta_1 Q_1 (E_1 - H_1))\right)}{\left(E_1^{\gamma} J_1 (a_2 + \delta_2 Q_2 (E_2 - H_2)) + E_2^{\gamma} J_2 (a_1 + \delta_1 Q_1 (E_1 - H_1))\right)^3} \tag{A10}$$

$$\frac{\partial W_1}{\partial Q_1} = -\frac{2E_1^{2\gamma}E_2^{\gamma}J_1^3 J_2 \delta_1 (E_1 - H_1)(a_2 + \delta_2 Q_2 (E_2 - H_2))}{\left(E_1^{\gamma} J_1 (a_2 + \delta_2 Q_2 (E_2 - H_2)) + E_2^{\gamma} J_2 (a_1 + \delta_1 Q_1 (E_1 - H_1))\right)^3} \tag{A11}$$

$$\frac{\partial W_2}{\partial Q_1} = \frac{2E_1^{\gamma}E_2^{2\gamma}J_1 J_2^3 \delta_1 (E_1 - H_1)(a_1 + \delta_1 Q_1 (E_1 - H_1))(a_2 + \delta_2 Q_2 (E_2 - H_2))}{\left(E_1^{\gamma} J_1 (a_2 + \delta_2 Q_2 (E_2 - H_2)) + E_2^{\gamma} J_2 (a_1 + \delta_1 Q_1 (E_1 - H_1))\right)^3} \tag{A12}$$

$$\frac{\partial f_1}{\partial \delta_1} = -\frac{2E_1^{\gamma}E_2^{2\gamma}J_1^2 J_2^2 Q_1 (E_1 - H_1)(a_2 + \delta_2 Q_2 (E_2 - H_2))}{\left(E_1^{\gamma} J_1 (a_2 + \delta_2 Q_2 (E_2 - H_2)) + E_2^{\gamma} J_2 (a_1 + \delta_1 Q_1 (E_1 - H_1))\right)^3} \tag{A13}$$

$$\frac{\partial f_2}{\partial \delta_1} = \frac{E_1^{\gamma}E_2^{\gamma}J_1 J_2^2 Q_1 (E_1 - H_1)\left(E_1^{\gamma} J_1 (a_2 + \delta_2 Q_2 (E_2 - H_2)) - E_2^{\gamma} J_2 (a_1 + \delta_1 Q_1 (E_1 - H_1))\right)}{\left(E_1^{\gamma} J_1 (a_2 + \delta_2 Q_2 (E_2 - H_2)) + E_2^{\gamma} J_2 (a_1 + \delta_1 Q_1 (E_1 - H_1))\right)^3} \tag{A14}$$

$$\frac{\partial W_1}{\partial \delta_1} = -\frac{2E_1^{2\gamma}E_2^{\gamma}J_1^3 J_2 Q_1(E_1 - H_1)(a_2 + \delta_2 Q_2(E_2 - H_2))}{\left(E_1^{\gamma}J_1(a_2 + \delta_2 Q_2(E_2 - H_2)) + E_2^{\gamma}J_2(a_1 + \delta_1 Q_1(E_1 - H_1))\right)^3} \tag{A15}$$

$$\frac{\partial W_2}{\partial \delta_1} = \frac{2E_1^{\gamma}E_2^{2\gamma}J_1 J_2^3 Q_1(E_1 - H_1)(a_1 + \delta_1 Q_1(E_1 - H_1))(a_2 + \delta_2 Q_2(E_2 - H_2))}{\left(E_1^{\gamma}J_1(a_2 + \delta_2 Q_2(E_2 - H_2)) + E_2^{\gamma}J_2(a_1 + \delta_1 Q_1(E_1 - H_1))\right)^3} \tag{A16}$$

Equations (A17), (A18), (A19), (A20) assume $\gamma = 1$.

$$\frac{\partial f_1}{\partial E_1} = -\frac{E_2 J_1^2 J_2(a_2 + \delta_2 Q_2(E_2 - H_2))(E_1 J_1(a_2 + \delta_2 Q_2(E_2 - H_2)) - E_2 J_2(a_1 - \delta_1 Q_1(E_1 + H_1)))}{(E_1 J_1(a_2 + \delta_2 Q_2(E_2 - H_2)) + E_2 J_2(a_1 + \delta_1 Q_1(E_1 - H_1)))^3} \tag{A17}$$

$$\frac{\partial f_2}{\partial E_1} = \frac{E_2 J_1 J_2^2(a_1 - \delta_1 Q_1 H_1)(E_2 J_2(a_1 + \delta_1 Q_1(E_1 - H_1)) - E_1 J_1(a_2 + \delta_2 Q_2(E_2 - H_2)))}{(E_1 J_1(a_2 + \delta_2 Q_2(E_2 - H_2)) + E_2 J_2(a_1 + \delta_1 Q_1(E_1 - H_1)))^3} \tag{A18}$$

$$\frac{\partial W_1}{\partial E_1} = \frac{2E_1 E_2 J_1^3 J_2(a_1 - \delta_1 Q_1 H_1)(a_2 + \delta_2 Q_2(E_2 - H_2))^2}{(E_1 J_1(a_2 + \delta_2 Q_2(E_2 - H_2)) + E_2 J_2(a_1 + \delta_1 Q_1(E_1 - H_1)))^3} \tag{A19}$$

$$\frac{\partial W_2}{\partial E_1} = -\frac{2E_2^2 J_1 J_2^3(a_1 - \delta_1 Q_1 H_1)(a_1 + \delta_1 Q_1(E_1 - H_1))(a_2 + \delta_2 Q_2(E_2 - H_2))}{(E_1 J_1(a_2 + \delta_2 Q_2(E_2 - H_2)) + E_2 J_2(a_1 + \delta_1 Q_1(E_1 - H_1)))^3} \tag{A20}$$

Equations (A21), (A22), (A23), (A24) assume $\gamma = 1$.

$$\frac{\partial f_1}{\partial H_1} = \frac{2E_1^{\gamma}E_2^{2\gamma}J_1^2 J_2^2 \delta_1 Q_1(a_2 + \delta_2 Q_2(E_2 - H_2))}{\left(E_1^{\gamma}J_1(a_2 + \delta_2 Q_2(E_2 - H_2)) + E_2^{\gamma}J_2(a_1 + \delta_1 Q_1(E_1 - H_1))\right)^3} \tag{A21}$$

$$\frac{\partial f_2}{\partial H_1} = \frac{E_1^{\gamma}E_2^{\gamma}J_1 J_2^2 \delta_1 Q_1\left(E_2^{\gamma}J_2(a_1 + \delta_1 Q_1(E_1 - H_1)) - E_1^{\gamma}J_1(a_2 + \delta_2 Q_2(E_2 - H_2))\right)}{\left(E_1^{\gamma}J_1(a_2 + \delta_2 Q_2(E_2 - H_2)) + E_2^{\gamma}J_2(a_1 + \delta_1 Q_1(E_1 - H_1))\right)^3} \tag{A22}$$

$$\frac{\partial W_1}{\partial H_1} = \frac{2E_1^{2\gamma}E_2^{\gamma}J_1^3 J_2 \delta_1 Q_1(a_2 + \delta_2 Q_2(E_2 - H_2))^2}{\left(E_1^{\gamma}J_1(a_2 + \delta_2 Q_2(E_2 - H_2)) + E_2^{\gamma}J_2(a_1 + \delta_1 Q_1(E_1 - H_1))\right)^3} \tag{A23}$$

$$\frac{\partial W_2}{\partial H_1} = -\frac{2E_1^{\gamma}E_2^{2\gamma}J_1 J_2^3 \delta_1 Q_1(a_1 + \delta_1 Q_1(E_1 - H_1))(a_2 + \delta_2 Q_2(E_2 - H_2))}{\left(E_1^{\gamma}J_1(a_2 + \delta_2 Q_2(E_2 - H_2)) + E_2^{\gamma}J_2(a_1 + \delta_1 Q_1(E_1 - H_1))\right)^3} \tag{A24}$$

$$\frac{\partial f_1}{\partial \alpha} = \frac{E_1^{\gamma}E_2^{\gamma}J_1^{\alpha+1}J_2^{\alpha}(a_1 + \delta_1 Q_1(E_1 - H_1))^{\alpha-1}(a_2 + \delta_2 Q_2(E_2 - H_2))^{\alpha} \times \left( \begin{array}{l} E_1^{\gamma}J_1^{\alpha}(a_2 + \delta_2 Q_2(E_2 - H_2))^{\alpha}\left(1 + \alpha Ln\left(\frac{J_2(a_1 + \delta_1 Q_1(E_1 - H_1))}{J_1(a_2 + \delta_2 Q_2(E_2 - H_2))}\right)\right) \\ + E_2^{\gamma}J_2^{\alpha}(a_1 + \delta_1 Q_1(E_1 - H_1))^{\alpha}\left(1 + \alpha Ln\left(\frac{J_1(a_2 + \delta_2 Q_2(E_2 - H_2))}{J_2(a_1 + \delta_1 Q_1(E_1 - H_1))}\right)\right) \end{array} \right)}{\left(E_1^{\gamma}J_1^{\alpha}(a_2 + \delta_2 Q_2(E_2 - H_2))^{\alpha} + E_2^{\gamma}J_2^{\alpha}(a_1 + \delta_1 Q_1(E_1 - H_1))^{\alpha}\right)^3} \tag{A25}$$

$$\frac{\partial f_2}{\partial \alpha} = \frac{E_1^{\gamma}E_2^{\gamma}J_1^{\alpha}J_2^{\alpha+1}(a_1 + \delta_1 Q_1(E_1 - H_1))^{\alpha}(a_2 + \delta_2 Q_2(E_2 - H_2))^{\alpha-1} \times \left( \begin{array}{l} E_1^{\gamma}J_1^{\alpha}(a_2 + \delta_2 Q_2(E_2 - H_2))^{\alpha}\left(1 + \alpha Ln\left(\frac{J_2(a_1 + \delta_1 Q_1(E_1 - H_1))}{J_1(a_2 + \delta_2 Q_2(E_2 - H_2))}\right)\right) \\ + E_2^{\gamma}J_2^{\alpha}(a_1 + \delta_1 Q_1(E_1 - H_1))^{\alpha}\left(1 + \alpha Ln\left(\frac{J_1(a_2 + \delta_2 Q_2(E_2 - H_2))}{J_2(a_1 + \delta_1 Q_1(E_1 - H_1))}\right)\right) \end{array} \right)}{\left(E_1^{\gamma}J_1^{\alpha}(a_2 + \delta_2 Q_2(E_2 - H_2))^{\alpha} + E_2^{\gamma}J_2^{\alpha}(a_1 + \delta_1 Q_1(E_1 - H_1))^{\alpha}\right)^3} \tag{A26}$$

$$\frac{\partial W_1}{\partial \alpha} = \frac{-E_1^\gamma E_2^\gamma J_1^{\alpha+1} J_2^\alpha (a_1 + \delta_1 Q_1 (E_1 - H_1))^\alpha (a_2 + \delta_2 Q_2 (E_2 - H_2))^\alpha \times \left( \begin{array}{c} \times \{ E_1^\gamma J_1^\alpha (a_2 + \delta_2 Q_2 (E_2 - H_2))^\alpha + E_2^\gamma J_2^\alpha (a_1 + \delta_1 Q_1 (E_1 - H_1))^\alpha \\ + \left( E_1^\gamma J_1^\alpha (1+\alpha)(a_2 + \delta_2 Q_2 (E_2 - H_2))^\alpha + E_2^\gamma J_2^\alpha (1-\alpha)(a_1 + \delta_1 Q_1 (E_1 - H_1))^\alpha \right) \\ \times Ln \left( \frac{J_2 (a_1 + \delta_1 Q_1 (E_1 - H_1))}{J_1 (a_2 + \delta_2 Q_2 (E_2 - H_2))} \right) \} \end{array} \right)}{\left( E_1^\gamma J_1^\alpha (a_2 + \delta_2 Q_2 (E_2 - H_2))^\alpha + E_2^\gamma J_2^\alpha (a_1 + \delta_1 Q_1 (E_1 - H_1))^\alpha \right)^3} \tag{A27}$$

$$\frac{\partial W_2}{\partial \alpha} = \frac{-E_1^\gamma E_2^\gamma J_1^\alpha J_2^{\alpha+1} (a_1 + \delta_1 Q_1 (E_1 - H_1))^\alpha (a_2 + \delta_2 Q_2 (E_2 - H_2))^\alpha \times \left( \begin{array}{c} \times \{ E_1^\gamma J_1^\alpha (a_2 + \delta_2 Q_2 (E_2 - H_2))^\alpha + E_2^\gamma J_2^\alpha (a_1 + \delta_1 Q_1 (E_1 - H_1))^\alpha \\ + \left( E_1^\gamma J_1^\alpha (1-\alpha)(a_2 + \delta_2 Q_2 (E_2 - H_2))^\alpha + E_2^\gamma J_2^\alpha (1+\alpha)(a_1 + \delta_1 Q_1 (E_1 - H_1))^\alpha \right) \\ \times Ln \left( \frac{J_1 (a_2 + \delta_2 Q_2 (E_2 - H_2))}{J_2 (a_1 + \delta_1 Q_1 (E_1 - H_1))} \right) \} \end{array} \right)}{\left( E_1^\gamma J_1^\alpha (a_2 + \delta_2 Q_2 (E_2 - H_2))^\alpha + E_2^\gamma J_2^\alpha (a_1 + \delta_1 Q_1 (E_1 - H_1))^\alpha \right)^3} \tag{A28}$$

$$\frac{\partial f_1}{\partial \gamma} = \frac{E_1^\gamma E_2^\gamma J_1^2 J_2 (a_2 + \delta_2 Q_2 (E_2 - H_2)) \times \left( E_2^\gamma J_2 (a_1 + \delta_1 Q_1 (E_1 - H_1)) - E_1^\gamma J_1 (a_2 + \delta_2 Q_2 (E_2 - H_2)) \right) (Ln(E_1) - Ln(E_2))}{\left( E_1^\gamma J_1 (a_2 + \delta_2 Q_2 (E_2 - H_2)) + E_2^\gamma J_2 (a_1 + \delta_1 Q_1 (E_1 - H_1)) \right)^3} \tag{A29}$$

$$\frac{\partial f_2}{\partial \gamma} = \frac{E_1^\gamma E_2^\gamma J_1 J_2^2 (a_1 + \delta_1 Q_1 (E_1 - H_1)) \times \left( E_2^\gamma J_2 (a_1 + \delta_1 Q_1 (E_1 - H_1)) - E_1^\gamma J_1 (a_2 + \delta_2 Q_2 (E_2 - H_2)) \right) (Ln(E_1) - Ln(E_2))}{\left( E_1^\gamma J_1 (a_2 + \delta_2 Q_2 (E_2 - H_2)) + E_2^\gamma J_2 (a_1 + \delta_1 Q_1 (E_1 - H_1)) \right)^3} \tag{A30}$$

$$\frac{\partial W_1}{\partial \gamma} = \frac{2 E_1^{2\gamma} E_2^\gamma J_1^3 J_2 (a_1 + \delta_1 Q_1 (E_1 - H_1))(a_2 + \delta_2 Q_2 (E_2 - H_2))^2 (Ln(E_1) - Ln(E_2))}{\left( E_1^\gamma J_1 (a_2 + \delta_2 Q_2 (E_2 - H_2)) + E_2^\gamma J_2 (a_1 + \delta_1 Q_1 (E_1 - H_1)) \right)^3} \tag{A31}$$

$$\frac{\partial W_2}{\partial \gamma} = - \frac{2 E_1^\gamma E_2^{2\gamma} J_1 J_2^3 (a_1 + \delta_1 Q_1 (E_1 - H_1))^2 (a_2 + \delta_2 Q_2 (E_2 - H_2))(Ln(E_1) - Ln(E_2))}{\left( E_1^\gamma J_1 (a_2 + \delta_2 Q_2 (E_2 - H_2)) + E_2^\gamma J_2 (a_1 + \delta_1 Q_1 (E_1 - H_1)) \right)^3} \tag{A32}$$

**Appendix C. Supplement**

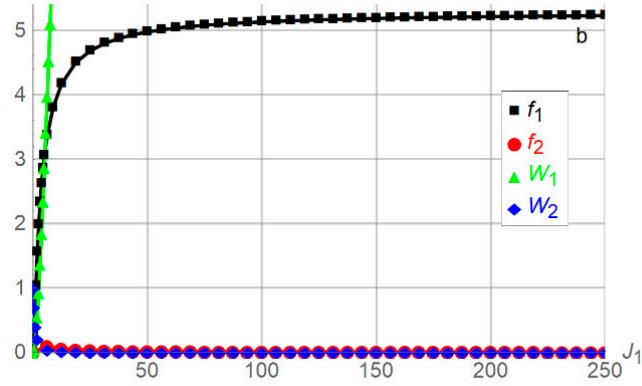

**Figure A1.** Supplementing Figure 2 panel b to illustrate $\lim_{J_1 \infty} f_1 = \frac{0.9}{0.17} \approx 5.29$, the plotranges along the horizontal and vertical axes are extended. Plotted are the media manipulation efforts $f_1$ and $f_2$ and the expected utilities $W_1$ and $W_2$ for players 1 and 2 as functions of $J_1$ relative to the benchmark parameter values $a_i = J_i = Q_i = \alpha = \gamma = \delta_i = 1$, $E_1 = 0.1$, $H_1 = 0.8$, which imply $E_2 = 0.9$, $H_2 = 0.2$, $i = 1, 2$.

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
