# Peer review of "A Game Theoretic Model of Adversaries and Media Manipulation"

_games, doi:10.3390/g10040048_

Round 1

Reviewer 1 Report

I found the paper moderately interested. The game is clearly laid out and the discussion of the simulated benchmark results was straightforward.

I did not detect any substantive problems with the proofs or explication.

There were a number of problems with the writing in the introduction and conclusion. First, awkward phrasing and word choice (use of plethora 2x, for example) made it hard to read through.

Second, the game felt unmotivated and the implications under-considered. To be a paper of more appeal, a concrete scenario should be added and discussed in the Introduction, during the simulations, and in the conclusion.

Finally, any connection to the literature is currently confined exclusively to the introduction, and thus feels tacked on as a motivation for the game, without reference to specific modelling choices made by the authors. This makes it hard to discern, in the reading, what precisely is the contribution to the literature. Weaving in references to other literature in the model section, and drawing attention to contrasts with existing literature in the conclusion both would help the reader understand the import of the work.

Author Response

Please see the attached listed response.

Reviewer 2 Report

This manuscript proposes a theoretical model for the game where two controversial actors (adversarially) interact with each other and then two players (e.g., media companies) choose their favorite actor (i.e. to support one actor out of ideological purpose). Players exert efforts (f1, f2) to report on the actors using some early-phase evidence (E1, E2). After players have conducted their efforts, full evidence (H1, H2) may become available. Each of the two players tries to make efforts such that its final utility W1, W2 (e.g., ideological goal) is optimized in the full phase of the game.

As one important contribution, this paper’s model can capture the phenomenon of how the change/difference from early evidence to full evidence may affect the strategic effort and the full-phased utility of both players. To establish this model, in addition to effort (f1, f2), early-phase evidence (E1, E2) and full evidence (H1, H2), the proposed utility equation in equation 5 involves many other parameters, such as contest intensity \alpha, stake J that players have in the interaction, early evidence ratio intensity \gamma, time discount parameter \delta, effort unit cost a, etc.

As another contribution, the authors thoroughly derived all the closed-form equation of the derivative of W and f w.r.t. each of the parameters they introduced. Upon these theoretical results, they achieve several property of the model regarding the conditions of when the derivative remains positive/negative.

Finally, authors conducted on numerical example in section5 to illustrate how f1, f2, W1, W2 will change w.r.t. the change in one of the parameters.

This paper is self-contained and well-written/well-structured. This paper can be improved if the following issues can be addressed or the answers are answered.

- Figure1: should change to “H1+H2=1, 0<=H1,H2<=1” for Actor1 and Actor2.

- Property 1-6: Please clarify and identify in main text that which equation(s) in appendix proves the corresponding property.

- Figure2b: Please add one more figure in appendix to help visualize your claim, lim_{J1->\infinite} f_1 = 5.29. Maybe larger y-lim and x-lim could help.

- The conclusion should be revised as it currently spends too many words repeating/describing the influence of each parameter on effort and utility. The authors should brief this part, and should 1) discuss how the framework applies to real word and/or 2) propose future work like generalizing the model to multi-players and multi-actor cases. 

Author Response

(The authors gave the same response as above.)
